# Investigation of a Large Kindred Reveals Cardiac Calsequestrin (*CASQ2*) as a Cause of Brugada Syndrome

**DOI:** 10.3390/genes15070822

**Published:** 2024-06-21

**Authors:** Maria d’Apolito, Francesco Santoro, Alessandra Ranaldi, Ilaria Ragnatela, Anna Laura Colia, Sara Cannito, Alessandra Margaglione, Girolamo D’Arienzo, Giovanna D’Andrea, PierLuigi Pellegrino, Rosa Santacroce, Natale Daniele Brunetti, Maurizio Margaglione

**Affiliations:** 1Medical Genetics, Department of Clinical and Experimental Medicine, University of Foggia, 71122 Foggia, Italy; alessandra.ranaldi@unifg.it (A.R.); annalaura.colia@unifg.it (A.L.C.);; 2Cardiology Unit, Department of Medical and Surgery Sciences, University of Foggia, 71122 Foggia, Italy

**Keywords:** Brugada syndrome, genetics, gene, *CASQ2*

## Abstract

Background: Brugada syndrome (BrS) is an inherited primary channelopathy syndrome associated with the risk of ventricular fibrillation (VF) and sudden cardiac death in a structurally normal heart. Aim of the Study: The aim of this study was to clinically and genetically evaluate a large family with severe autosomal dominant Brugada syndrome. Methods: Clinical and genetic studies were performed. Genetic analysis was conducted with NGS technologies (WES) using the Illumina instrument. According to the standard procedure, variants found by WES were confirmed in all available families by Sanger sequencing. The effect of the variants was studied by using in silico prediction of pathogenicity. Results: The proband was a 52-year-old man who was admitted to the emergency department for syncope at rest. WES of the index case identified a heterozygous VUS *CASQ2*, c.532T>C, p.(Tyr178His). We studied the segregation of the variation in all pedigree members. All the patients were heterozygous for the variation *CASQ2* p.(Tyr178His), whereas the remaining healthy individuals in the family were homozygous for the normal allele. Structural analysis of *CASQ2* p.(Tyr178His) was performed and revealed an important effect of the missense variation on monomer stability. The *CASQ2* Tyr180 residue is located inside the sarcoplasmic reticulum (SR) junctional face membrane interaction domain and is predicted to disrupt filamentation. Conclusions: Our data suggest that the p.Tyr178His substitution is associated with BrS in the family investigated, affecting the stability of the protein, disrupting filamentation at the interdimer interface, and affecting the subsequent formation of tetramers and polymers that contain calcium-binding sites.

## 1. Introduction

Brugada syndrome (BrS) is one of the main hereditary channelopathies characterized by risk of ventricular fibrillation (VF) and sudden cardiac death in an anatomically healthy heart. BrS was first described by Pedro and Josep Brugada in 1992 as a hereditary arrhythmogenic disorder characterized by clinical–electrocardiographic arrhythmia, with a low prevalence globally (0.5 per 1000 or 5 to 20 per 10,000 individuals) [1,2,3]. BrS is found predominantly in men aged between 30 and 40, with a male/female ratio of 9:1 in Southeast Asia and 3:1 among Caucasians [4].

BrS is characterized by the presence of ST segment elevation in the right precordial leads (V1 to V3), referred to as electrocardiogram (ECG) type I. The diagnosis is established on the existence of spontaneous or drug-induced ST segment elevation characterized by ≥2 mm elevation of the J-point and ST segment, either superiorly convex “arched” (BrS type II) or descending linear (BrS type III).

The ST elevation is followed by a symmetrical negative T wave in ≥1 right and/or high right precordial leads [5,6].

In spite of the most recent models on further inheritance pathways, BrS is still considered to be an autosomal dominant Mendelian disorder inherited with incomplete penetrance. Genetic mutations have been identified in 11–28% of patients with BrS, with a major percentage affecting the *SCN5A* (sodium voltage-gated channel alpha subunit 5) gene [7]. Actually, *SCN5A* is considered the only clinically relevant gene evaluated, even if it is mutated in only about 20% of patients with BrS [7,8]. Pathogenic variations in the *SCN5A* gene, which encodes the α subunit of the voltage-gated cardiac Na+ channel protein (Nav1.5), were detected in patients with BrS, impairing the proper function of the channel. Genetic variants in over 27 other genes have also been associated with the pathophysiology of BrS. Although numerous pathogenic gene variants have been demonstrated to modify the classic functions of potassium with Gain-of-Function (↑GOF) and sodium with Loss-of-Function (↓LOF), the role of all genes, excluding *SCN5A,* has currently been disputed [8]. Pathogenic variations in genes coding for potassium and calcium channels have also been reported. All other genes that account for less than 10% of cases are considered as minority genes with potential implications for BrS or BrS-like phenotypes. Polymorphisms and non-hereditary factors such as fever are thought to impact disease manifestation [9].

Our team recently discovered a novel *DLG1* variant in the BrS family. Mutations in the *DLG1* gene may alter the structure of the multichannel protein complex, resulting in structural changes that affect the protein’s function and its ability to regulate ion channels [10]. The introduction of new NGS technologies, whole-exome and whole-genome sequencing, allowed the identification of new genes by analyzing WES data from patients with comparable clinical characteristics with high accuracy and reduced cost. These techniques can also be used to discover alterations in genes that have been previously recognized to cause disease.

We report the findings of clinical and genetic studies conducted in an Italian family with BrS. Whole-exome sequencing (WES) was used in genetic testing on the Illumina platform.

## 2. Materials and Methods

### 2.1. Clinical and Genetic Study

The index patient, a 52-year-old Italian man, was diagnosed with Brugada syndrome at the age of 46. A thorough medical and cardiovascular examination was performed on every family member who was available for evaluation. Written informed consent was obtained for both clinical and genetic analysis. Medical and genetic examinations were carried out in compliance with the Helsinki Declaration. For subjects under the age of eighteen, written informed consent was acquired from a legal representative. The manuscript’s data were all appropriately anonymized. The local ethical committee gave its approval to the study (code of protocol: 3261/CE/20).

### 2.2. Genes of Interest

The Appendix A contains the gene list for the arrhythmic sudden death syndrome that was investigated. The gene list was expanded through a search of the current literature and the Human Gene Mutation Database (HGMD). It includes candidate genes for arrhythmias such as Brugada syndrome and other well-known arrhythmias, in addition to arrhythmic cardiomyopathy gene in vivo models (Appendix A).

### 2.3. Whole-Exome Sequencing

WES and analysis were carried out as previously reported above [10]. Briefly, DNA was extracted from whole blood samples using the automated extraction instrument MagCore^®^ Plus II according to the manufacturer’s instructions. To capture all coding regions and exon–intron boundaries (±50 bps), enrichment was carried out using Illumina DNA prep with enrichment (Illumina, San Diego, CA, USA). Paired-end sequencing was then performed with an Illumina NextSeq 550 (Illumina, San Diego, CA, USA). Following that, the Genome Analysis Toolkit (GATK 1.6) was used to process the raw data. Using the BaseSpace Variant Interpreter software, Version 2.17.0.60 (Illumina, San Diego, CA, USA), the reads were aligned to the human genome (GRCh37), and variant calling was completed [10].

Several strategies have been used to reduce the quantity of possibly damaging genetic defects: (1) selecting and eliminating variants with a quality score of more than 30; (2) excluding variants with a MAF > 0.01 in the Single Nucleotide Polymorphism Database (https://www.ncbi.nlm.nih.gov/snp/) (accessed on 20 March 2024), Exome Aggregation Consortium (https://gnomad.broadinstitute.org/) (accessed on 20 March 2024), Exome Variant Server (https://www.ebi.ac.uk/eva) (accessed on 20 March 2024), 1000 Genomes Project (https://www.internationalgenome.org/1000-genomes-browsers) (accessed on 20 March 2024), and research publications; (3) excluding variants in intron regions or synonymous variants; (4) choosing variants that segregate according to the hypothetical pattern of inheritance; and (5) searching databases, such as ClinVar (http://www.ncbi.nlm.nih.gov/clinvar/) (accessed on 20 March 2024), OMIM (http://www.omim.org) (accessed on 20 March 2024), and the Human Gene Mutation Database locus-specific database (https://www.hgvs.org/) (accessed on 20 March 2024) to additionally select the candidate gene mutation. Approximately 531 variants were retained following the functional annotation step, which was carried out in agreement with the impact on protein function and pre-existing information on phenotype.

### 2.4. Segregation Analysis

Using standard procedures, Sanger sequencing was performed for the segregation analysis. In summary, the target region was amplified using specific 20-base primers by PCR amplification (CTNNA3_F 5′AAACACAGACACACGCAAAA3′; CTNNA3_R 5′ TTGTGCAGCTGTTATTGGCA 3′; CASQ2_F 5′TCCCTCCAATGCACTCTGTT3′, CASQ2_R 5′TTTAGACTGATCGGCTGGGG 3′). BigDyeTM Terminator V1.3 (Thermo Fisher Scientific, Waltham, MA, USA) was used to perform cycle sequencing on the PCR products. This was followed by BigDye XTerminatorTM Purification (Applied Biosystems, Waltham, MA, USA) and capillary sequencing on a SeqStudio (Thermo Fisher Scientific, Waltham, MA, USA) [11].

### 2.5. In Silico Study and Protein Structure Modeling

We performed functional annotation based on prior data on protein function impact and phenotype to further select candidate gene variants. To determine the degree of potential impact of a variant on protein function, an in silico pathogenicity estimation was carried out with several bioinformatic tools for nonsynonymous amino acid variation. The NCBI GenBank accession numbers NM_001232.4 (*CASQ2*) and NM_013266.4 (*CTNNA3*) and UniProt Identifiers O14958 (*CASQ2*) and Q9UI47(*CTNNA3*) were used as reference sequences. The effect of the *CTNNA3* (p.Cys521Arg) and *CASQ2* (p.Tyr178His) substitutions was investigated with an in silico pathogenicity prediction web server, as well as Polyphen-2 (http://genetics.bwh.harvard.edu/pph2/, accessed on 20 March 2024), SIFT (https://bio.tools/sift, accessed on 20 March 2024), Mutation Taster (http://www.mutationtaster.org/, accessed on 20 March 2024), PROVEAN (http://provean.jcvi.org/index.php, accessed on 20 March 2024), REVEL (https://sites.google.com/site/revelgenomics/, accessed on 20 March 2024), and Mutation Assessor (http://mutationassessor.org/r3/, accessed on 20 March 2024).

PRIDE (Proteomics IDEntification Database) Project PXD012174 (https://www.ebi.ac.uk/pride/archive/projects/PXD012174 (accessed on 10 May 2024)) was used to assess the possibility of the effect of the mutation on the secondary structure using proteomics information, as well as protein and peptide identifications, post-translational modifications, and the human phosphoproteome map.

DynaMut2 was then utilized to explore the potential deleterious impact of the missense variation by using the crystal structure of a human cardiac calsequestrin filament (60WV PDB) as a template.

DynaMut2 (https://biosig.lab.uq.edu.au/dynamut2/ (accessed on 10 May 2024)), a web server, can be used to evaluate the effects of mutations on the vibrational entropy variations induced and modifications in protein dynamics and stability, as well as to analyze and visualize protein dynamics by sampling conformations [12].

The wild-type and mutant proteins of the CASQ2 molecular models were visualized using Swiss-PdbViewer version 4.1.0.

## 3. Results

### 3.1. Proband Patient Characteristics and Pedigree Investigation

The index case was a 52-year-old man who was admitted to the emergency department because of syncope at rest. The 12-lead rest electrocardiogram (ECG) revealed a type 1 Brugada pattern and left anterior hemiblock (Figure 1). The patient had a family history of unexplained sudden cardiac death (his sister died at rest at the age of three, II-3). The proband also had a history of chronic gastritis in addition to cardiac lesions. A single-chamber cardiac defibrillator was implanted while the patient was in the hospital. After two years, the patient experienced a single appropriate DC shock due to ventricular tachycardia (Torsade de pointes) degenerating into ventricular fibrillation.

The proband’s sons (III-5, III-6) all had a BrS type 1 ECG drug-induced pattern. None of them had a history of syncope and none experienced cardiac arrhythmias at ECG-Holter. Therefore, both of them were under cardiological controls. Evaluation of the proband’s first-degree family identified two additional family members (II-7, II-12) with a spontaneous type I BrS ECG pattern (II-12) and one with a BrS type 1 ECG drug-induced pattern with flecainide testing (II-7). Concerning the proband’s second-degree relatives, there were five additional family members (III-5, III-6, III-7, III-9, III-14). Two presented with a spontaneous BrS type 1 ECG pattern (III-7, III-14) and three with a BrS type 1 ECG drug-induced pattern (III-5, III-6, III-9). A patient with a history of syncope and spontaneous type 1 BrS ECG underwent ICD implant (III-7) (Table 1).

### 3.2. Genetic Analysis Results and Prioritization of Variants

The exon sequencing of the proband produced a large number of variants (11,878). Due to the fact that the Brugada syndrome is autosomal dominant, we focused our filtering on heterozygous variants. The inclusion of variants with a heterozygous allele frequency ≤1% revealed 531 variants with a MAF < 0.01. The analysis of variants produced by the free software BaseSpace Variant Interpreter (Illumina, San Diego, CA, USA) was limited to virtual subpanels by applying a gene list linked to the syndrome (see Appendix A).

Only 21 variations were left after this step. Variations were categorized (i.e., exonic, intronic, and untranslated regions; synonymous, nonsynonymous, stop gain/loss, frameshift) and were annotated according ACMG classification [13]. The index case (II-4) had no pathogenic variant (Appendix A). Seven synonymous variants were eliminated because they were annotated as benign or probably benign and located in any gene not associated with BrS to date. The other 14 nonsynonymous, frameshift deletion, and exonic variants were annotated using various public databases: VarSome, Ensemble, gnomAD (exomes), gnomAD (genomes). Five of these were classified as benign or probably benign and nine as variants of uncertain significance (VUSs). No VUSs in other known Brugada syndrome-associated genes were found. Only two heterozygous VUSs remained to be further investigated: *CTNNA3* (NM_013266.3) c.1561T>C, (p.Cys521Arg) and *CASQ2*, c.532T>C, (p.Tyr178His) (Figure 2, Table 2).

The *CTNNA3* gene coding α T-catenin 3 is associated with familial Arrhythmogenic Right Ventricular Dysplasia, 13 ARVC (OMIM ID#615616). The missense variant c.1561T>C; p.Cys521Arg replaces the amino acid cysteine with arginine at codon 521. This variant is not present in HGMD. In silico prediction tools predict the pathogenetic effect of this variant on the structure and function of the protein (Sift: deleterious (0.01); PolyPhen: probably damaging (0.955)). Its frequency in the Genome Aggregation Database (gnomAD) is Cƒ = 0.00000636 exosomes (cov 31.5) and not found in genomes (cov 32.2). The Conservation Score phyloP100 was 9.325 and represented a measure of evolutional constraint indicating evolutionary conservation. The information at our disposal did not allow us to attribute a certainly pathological significance to this variant; therefore, the variant was considered a VUS. The latest molecular studies propose that ARVC and Brugada syndrome are not fully distinct disorders, but there is partial clinical overlap between the two conditions in a continuum of sodium current deficit and structural change [14].

The analysis performed as described above allowed the identification of an additional variant in exon 4 of the *CASQ2* gene (NM_001232.3) c.532T>C (p.Tyr178His) in heterozygosity. Catecholaminergic Polymorphic Ventricular Tachycardia (CPVT) (OMIM ID#611938) is typically caused by a compound heterozygous or homozygous variation in the *CASQ2* gene. Furthermore, heterozygous mutations of the *CASQ2* gene have been described as a VUS in a patient with BrS [15] and in a young subject with a pharmacologically induced type 1 Brugada ECG pattern and Sotos syndrome [16]. However, the significance of this association has not been fully clarified.

*CASQ2* is located in the short arm of chromosome 1 and encodes cardiac calsequestrin, an intra-sarcoplasmic reticulum protein involved in the regulation of calcium ion levels in cardiac cells [17,18].

The variant p.Tyr178His (rs1648031031) in the *CASQ2* gene results from a T-to-G substitution at nucleotide position 532, which implicates the amino acid substitution of a Tyrosine with Histidine at position 178. The comparison of the CASQ protein with ortholog proteins showed a moderate conservation across different species (Figure 3C). The variant has a rare frequency Cƒ = 0.0000014 (gnomAD exomes) and is classified according to the ACMG criteria [18] as a variant of uncertain significance (VUS). It is annotated in ClinVar (RCV001170445).

### 3.3. Genetic Studies

We confirmed the changes in the *CASQ2* and *CTNNA3* genes detected by WES using PCR amplification with specific primers bordering the variant. The variation was validated by bi-directional Sanger sequencing, as shown in Figure 3A,B.

The segregation of variants among all available family members was then characterized. The *CTNNA3* p.Cys521Arg variant did not cosegregate with the phenotype and was declared benign. All of the patients were heterozygous for the *CASQ2* gene p.(Tyr178His), whereas the remainder of the unaffected family members were homozygous for the normal allele. Direct sequencing identified the *CASQ2* variant in all of the proband’s sons (III-5, III-6), who all had a BrS type 1 ECG drug-induced pattern. Evaluation of the index case’s first-degree family identified two other family members (II-7, II-12) carrying the *CASQ2* variant, one patient with a spontaneous type I BrS ECG pattern (II-12) and one with a BrS type 1 ECG drug-induced pattern with flecainide testing (II-7). Concerning the proband’s second-degree relatives, there were five additional family members (III-5, III-6, III-7, III-9, III-14) carrying the *CASQ2* variation. Two showed a spontaneous BrS type 1 ECG pattern (III-7, III-14) and three a BrS type 1 ECG drug-induced pattern (III-5, III-6, III-9). A patient with a history of syncope and a spontaneous type 1 BrS ECG underwent ICD implant (III-7). Ten negative noncarriers and seven *CASQ2* genotype-positive carriers were examined collectively. Among the carriers, four had a type 1 Brugada ECG (II-4, II-12, III-7, III-14) and four had a type 1 drug-induced Brugada ECG (II-7, III-5, III-6, III-9). The distribution of affected people in the pedigree suggests dominant inheritance (Figure 4).

### 3.4. In Silico Evaluation and Structural Analysis

In silico prediction tools predict that p.Tyr178His has a moderate impact on the protein’s structure and function. This substitution arises at a site that is not conserved across species, and Histidine has been found at this location throughout evolution. However, the alteration p.Tyr178His may have an impact on secondary protein structure as these residues differ in certain properties (Figure 3C).

To assess the possibility of the effect of the variant on the secondary structure we searched in PRIDE (Proteomics IDEntification Database) Project PXD012174 (https://www.ebi.ac.uk/pride/archive/projects/PXD012174 (accessed on 10 May 2024)). This tool identified Tyrosine 178 as a modified residue referred to as Phosphotyrosine. This project offers an open data repository for proteomics with protein and peptide identifications, post-translational modifications, and the human phosphoproteome map, using automatic assertion inferred from a combination of experimental and computational evidence.

To analyze the movement and flexibility of the mutant p.Tyr178His protein, we used the crystal structure of a human cardiac calsequestrin filament (CASQ2 PDB: 6OWV) as a template and DynaMut2, a web server that combines Normal Mode Analysis (NMA) methods. DynaMut2 predicted a significant effect of the missense p.Tyr178His variation on protein stability with a Predicted Stability Change (ΔΔGStability) of −0.11 kcal/mol, indicating a destabilizing effect. In this tool, ΔΔG ≥ 0 is considered stabilizing, and ΔΔG < 0 is considered destabilizing [19]. The modeling identifies modifications in a mutant structure, allowing the formation of new extra bonds between the mutated p.Tyr178His and residues located in the solvent-exposed domain (Figure 5).

*CASQ2* missense mutations previously associated with CPVT have been mapped to the recently reported crystal structure of the cardiac CASQ2 filament (Figure 6), and the supposed functional classes of the *CASQ2* missense variant have been described [19].

The locations of damaging *CASQ2* dominant p.S173I and p.K180R mutations were mapped and both localized to a solvent cavity formed by the interdimer interface. The *CASQ2* Tyr180 residue match the same surface, located in the sarcoplasmic reticulum’s (SR) junctional face membrane interaction domain. In contrast, the missense variants exclusively transmitted in an autosomal recessive manner localized to either the intradimer interface or the hydrophobic core.

## 4. Discussion

In the current study, a large Italian family with Brugada syndrome was investigated to detect the potential genetic cause responsible for Brugada syndrome, using a WES methodology. No variants associated with the other main susceptibility genes were identified; the index case was studied using WES to identify additional pathogenic variants in familiar arrhythmia candidate genes.

In the proband, we identified two rare missense variations classified as VUSs: *CTNNA3* (NM_013266.3) c.1561T>C, (p.Cys521Arg) and *CASQ2*, c.532T>C, (p.Tyr178His).

The *CTNNA3* p.Cys521Arg variant was classified as benign since it did not cosegregate with illness. The results from the segregation study identified a rare missense variant in exon 4 of the *CASQ2* gene (NM_001232.3) c.532T>C (p.Tyr178His) in heterozygosity. All the family members affected were heterozygous for the variation *CASQ2* gene (p.Tyr178His); the remaining healthy members of the family, however, were homozygous for the normal allele. Dominant inheritance is consistent with the pedigree’s affected individual distribution. Cardiac calsequestrin 2 gene (*CASQ2*) maps to chromosome 1p13.3-p11, which encodes 399 amino acids through 11 exons [20]. The pathophysiology of CPVT is associated with mutations in this gene, which impair the release of SR calcium [21,22]. All reported pathogenic variants are autosomal recessive, both homozygous and compound heterozygous [23,24,25,26]. Recently, two novel missense variants in *CASQ2* Lys180Arg and Ser173Ile have been described as autosomal dominant *CASQ2* variants that may cause CPVT [20]. For the first time, a heterozygous mutation of the *CASQ2* gene has been reported as a VUS in a patient with BrS [15]. Recently, a young individual with Sotos syndrome and a pharmacologically induced type 1 Brugada ECG pattern was shown to have a heterozygous mutation of *CASQ2* [16]. The implications of a *CASQ2* gene mutation in this latter case report are still unclear [16].

CASQ2 is an important intracellular calcium-storage protein involved in the regulation of the release of Ca^2+^ from the junctional sarcoplasmic reticulum (SR). CASQ2 plays an essential role in causing cardiac muscle contraction by regulating the release of luminal Ca^2+^ through the RYR2 channel. The CASQ2 protein colocalizes with RYR2, triadin, and junctin in the junctional SR [27,28]. In the heart, CASQ2 controls SR free Ca levels and Ca release during excitation–contraction coupling (E-C) through RYR2.

The proposed mechanisms of *CASQ2*-CPVT involve the loss of calcium buffering, altered regulation of RYR2 activity, and changes in the sarcoplasmic ultrastructure and its constituent proteins. Mutations in *CASQ2* cause a lack of control of the RYR2 channel and, consequently, a persistent release of Ca^2+^ into the cytoplasm, followed by arrhythmias.

The calcium-induced oligomerization of CASQ2 into filaments in the junctional SR facilitates high-capacity calcium storage [29,30]. Front-to-front dimerization of monomers and consequent back-to-back binding of dimers leads to the formation of tetramers and highly organized polymers. CASQ2 polymers contain calcium-binding sites inside electronegative pockets present at intradimer and interdimer interfaces. This process is known as oligomerization or filamentation [31,32]. CASQ2 polymerization and depolymerization play a key role in regulating RYR2 during the EC coupling cycle, affecting the end of Ca^2+^ release from the SR through the binding of its monomers to the RYR2 channel [33].

New information about the CASQ2 filament structure has been revealed by the recent X-ray crystallographic structure of 60 WV PDB (https://www.wwpdb.org/pdb?id=pdb_00006owv (accessed on 10 May 2024)). There is now a known biochemical mechanism explaining how dominant-acting calsequestrin mutations cause fatal arrhythmia [19]. Different mechanisms underlying the positive CPVT phenotype have been associated with a heterozygous CASQ2 variant but seem to be dependent on the physical position and function of specific residues in the CASQ2 filament. Unlike in recessive genetic models, these mutations impact filament formation at the interdimer interface, disrupting back-to-back dimer binding and preserving front-to-front dimerization of monomers [19]. Structural analysis of the new crystal structure 60WV PDB of cardiac calsequestrin filament revealed that the dominant variants *CASQ2* Lys180Arg and Ser173Ile, previously associated with CPVT, mapped to regions relevant to filamentation. Both missense variations localized to an electronegative solvent cavity involved in back-to-back binding required for polymerization [34]. Recently, it has been proposed that the Lys180Arg mutation could disturb the polymerization of CASQ2 [34] affecting the stability of the protein, disrupting filamentation at the interdimer interface, and subsequently forming tetramers and polymers containing calcium-binding sites. The polymerization of CASQ2 has a fundamental role in the capacity of CASQ2 to bind Ca^2+^ and to supply a structural component for the SR. Based on the findings of in vitro turbidity assays, the authors established that Lys180Arg and other autosomal dominant *CASQ2* mutations reduced the ability of CASQ2 to polymerize in the SR appropriately.

Recently, the CASQ2 polymer X-ray crystallographic structure also showed new potential Ca-binding sites in the polymer and a cavity within the center of the polymer where Ca ions could bind. It has been proposed that Lys180 localized in the interdimer region within the CASQ2 polymer. Two potential Ca-binding sites near Lys180 were also found in the same region. Because Lys180Arg is involved in the interdimer region, it could affect the ability of CASQ2 polymers to bind Ca. This mechanism could explain how Lys180Arg impacts the dynamic buffering capability of the SR [31].

The heterozygous variant identified in this study, *CASQ2* p.Tyr178His substitution, is located in the same solvent pocket created by the interdimer interface.

## 5. Conclusions

Our data suggest that the p.Tyr178His substitution is associated with dominant Brugada syndrome in the family investigated. We hypothesize that p.Tyr178His could cause a defective oligomerization, similarly to Lys180Arg, contributing to the loss of calcium buffering and altering of the sarcoplasmic ultrastructure and its constituent proteins.

In conclusion, mutant p.Tyr178His could contribute to an alteration in calcium homeostasis and represent a new pathophysiologic cause of autosomal dominant Brugada syndrome.

## Figures and Tables

**Figure 1 genes-15-00822-f001:**
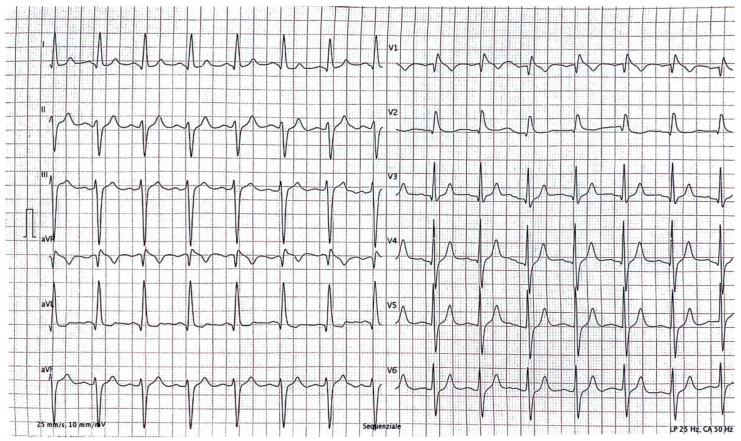
Rest 12-lead ECG of the proband presenting Brugada type 1 ECG pattern.

**Figure 2 genes-15-00822-f002:**
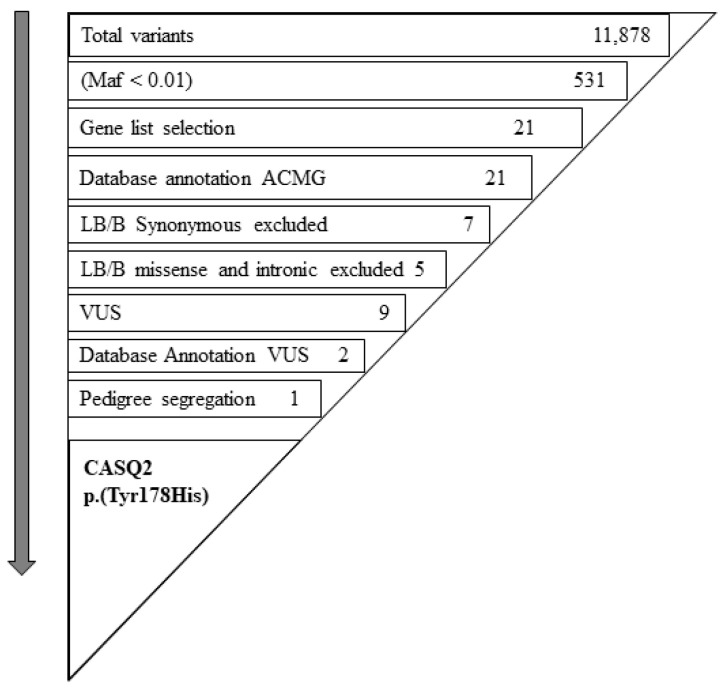
Prioritization of the missense variant p.Tyr178His in *CASQ2* gene.

**Figure 3 genes-15-00822-f003:**
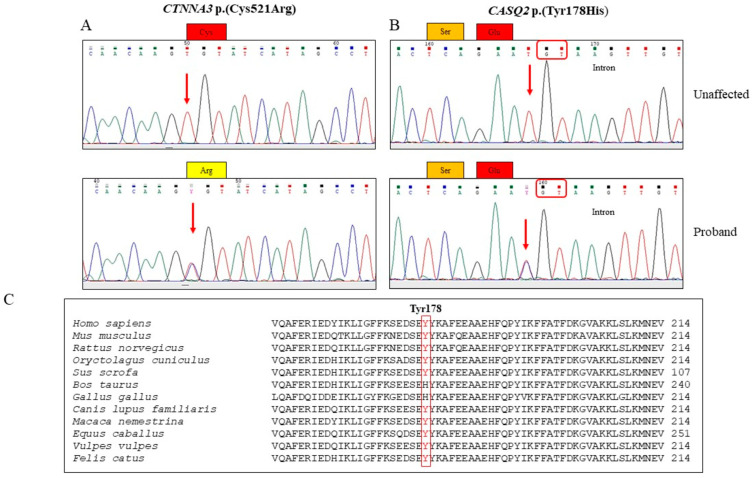
(**A**) Electropherograms representing the wild-type *CTNNA3* sequence (upper panel) identified in unaffected family members and the heterozygous C521R variant (lower panel) identified in the proband. (**B**) Electropherograms showing the wild-type *CASQ2* sequence (upper panel) identified in unaffected family members and the heterozygous *CASQ2* Tyr178His variant (lower panel) diagnosed in proband II-4 and family members II-7, II-12, III-5, III-6, III-7, III-9, and III-14. Donor splice site is highlighted in red. (**C**) Alignment of CASQ2 sequence (#178 amino acid site is highlighted in red) across species.

**Figure 4 genes-15-00822-f004:**
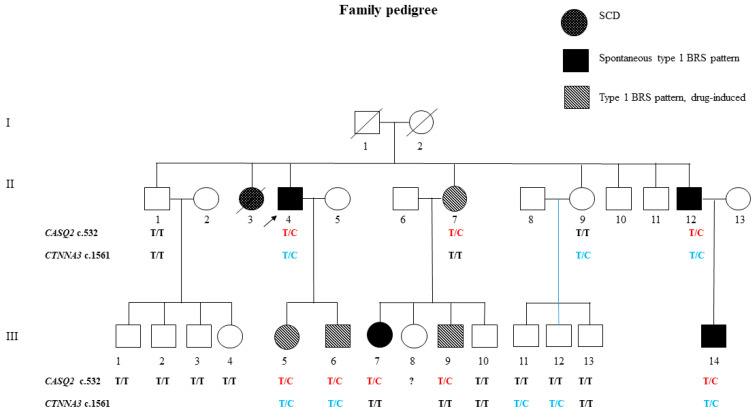
Family pedigree of the proband with the rare *CTNNA3* and *CASQ2* variations. Males are represented by squares and females by circles. Individuals with open symbols are asymptomatic, while those with filled symbols have the *CASQ2* variant. The clinical phenotype (BrS type 1 ECG and an arrhythmogenic syndrome) is indicated according to the trauma code. T/C: heterozygote for the *CASQ2* p.(Tyr178His) variant; T/T: noncarrier.

**Figure 5 genes-15-00822-f005:**
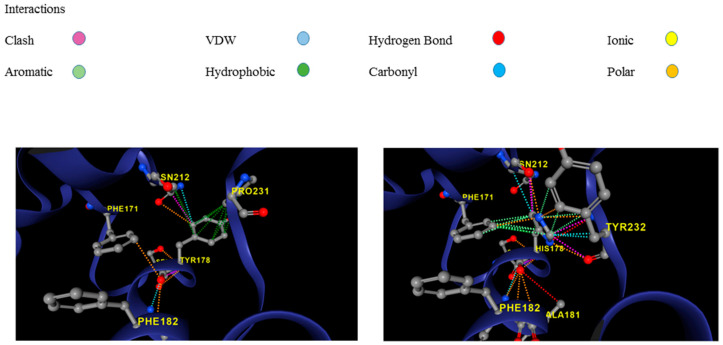
Using the DynaMut2 online server, the predicted structures of the wild-type (left: 178Y) and variant protein (right: 178H) are compared.

**Figure 6 genes-15-00822-f006:**
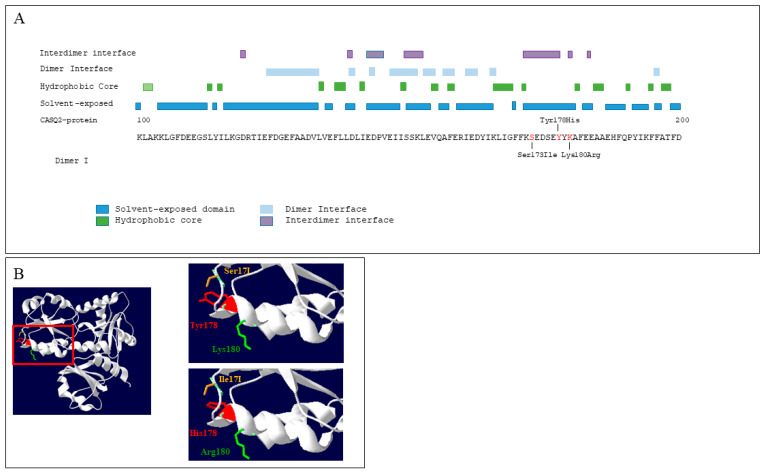
(**A**) CASQ2 partial protein sequence (aa 100–200), with amino acid allocation of Tyr 178, Ser173, and Lys180 within interdimer interface domain. (**B**) Swiss-PdbViewer was used to visualize the expected structures of both the wild-type (Y178) and mutant (H178) proteins in the interdimer domain. A recent X-ray crystallographic structure, 60 WV PDB, was used as a template.

**Table 1 genes-15-00822-t001:** Cardiac phenotypes of the members carrying the CASQ2 variant.

Patient	Sex	Age	History of Syncope	ECG Eatures	ICD Implant
II-4 proband	M	52	Yes	Spontaneous type 1 BrS pattern	+
II-7	F	60	No	Type 1 BrS pattern, drug-induced	-
II-12	M	44	No	Spontaneous type 1 BrS pattern	-
III-5	F	37	No	Type 1 BrS pattern, drug-induced	-
III-6	M	32	No	Type 1 BrS pattern, drug-induced	-
III-7	F	34	Yes	Spontaneous type 1 BrS pattern	+
III-9	M	31	No	Type 1 BrS pattern, drug-induced	-
III-14	M	14	No	Spontaneous type 1 BrS pattern	-

**Table 2 genes-15-00822-t002:** Exonic variants prioritized in the analysis of the Brugada proband. Exonic classification, allele frequencies, ACMG classification, and ClinVar annotation.

Gene	dbSNP	Variant	Amino Acid Change	Exonic Classification	Allele Frequency	ACMG Classification
*CTNNA3*	rs1228977390	c.1561T>C	p.(Cys521Arg)	missense variant	Exomes Cƒ = 0.00000636 (cov 31.5) Genomes not found (cov 32.2)	VUS RCV003145906
*CASQ2*	rs1648031031	c.532T>C	p.(Tyr178His)	missense variant	Exomes Cƒ = 0.0000014 (cov31.3). Genomes not found (cov 32.4)	VUS RCV001170445

## Data Availability

The original contributions presented in the study are included in the article/Appendix A, further inquiries can be directed to the corresponding author.

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
