# Peer review of "Investigation of a Large Kindred Reveals Cardiac Calsequestrin (CASQ2) as a Cause of Brugada Syndrome"

_genes, 2024, doi:10.3390/genes15070822_

Round 1

Reviewer 1 Report

Comments and Suggestions for Authors

Authors reported a new potential gene as cause of BrS in a family. Interesting case but several points should be clarified:

- Authors mention in Introduction that BrS is an oligogenic disease. It is not widely-accepted nowadays. Large part of cases showed a mendelian inheritance and clinical symptoms may be induced by external triggers.

- Authors mention that more than 20 genes have been associated with BrS. It is not correct. Today, only SCN5A is the single and main gene. All other genes are considered as minority genes with potential implication to BrS or BrS-like phenotypes.

- Why exclusion of rare synonymous variants? It is reported that some cases of BrS are due to synonymous rare variants.

- All rare variants classified as VUS should be segregated in the family, especially if located in any gene not associated with BrS to date.

- A table including genetic data of all rare variants (synonymous, missense, ...) could be of interest for readers. Any of these variants previously reported?

- Please, modify "mutation" to "rare variant" due to mutation concerns to any alteration in the genome.

- Genes (in all the manuscript) should be written in italic.

- Genes included in the table (S1) should be written in italic.

Comments on the Quality of English Language

Minor mistakes

Reviewer 2 Report

Comments and Suggestions for Authors

The manuscript by d’Apolito and coauthors reports the identification of a variant in Calsequestrin 2 as the cause of Brugada syndrome in a large family. Based on structural analysis, this variant was proposed to affect protein stability by disrupting filamentation that would be expected to impact subsequent formation of tetramers and polymers containing calcium-binding sites. This is an interesting finding and represents the first report directly implicating CASQ2 in Brugada syndrome.

1) On page 4 lines 165-176, the authors describe the clinical phenotype of various family members and also mention whether or not they carry the CASQ2 variant. Assuming that this is the actual variant that was identified by the WES analysis described in section 3.2, maybe the authors should consider including this paragraph at a later point in their manuscript along with the corresponding family tree.

2) Figure 2 should be enlarged in size as it is difficult to read the font. Also, in Figure 2C it is why Ser and Glu amino acids are indicated considering that the mutation is pTyr178His. Also what does the red circled bases depict?

3) A figure showing conservation of CASQ2 protein sequence around the variant could be useful

4)  Consistency on nomenclature would be advisable as the variant is referred to as either p.Tyr178His or Y178H at different points of the manuscript.

5) Figure 5B is unclear, especially regarding the observed changes in the presence of the Y178H variant. There is currently no reference to this part of the figure in the manuscript, the authors should include a brief description on this. Could the authors also explain what the predicted changes in filamentation would mean for the function of the protein, that is in relation to calcium binding/ calcium homeostasis and SR calcium release through RyR.

6) Based on the position of the CASQ2 variant, the authors propose that it affects protein oligomerization. Could this variant impact CASQ2 protein interactions with the RYR2-protein complex? Please comment

Comments on the Quality of English Language

Minor editing is required

Round 2

Reviewer 1 Report

Comments and Suggestions for Authors

Most of points have been performed. Please check again genes in italic (mainly in new sentences added in red color).

Author Response

We apologize to the reviewer. 'Genes' has been written in Italic in all manuscript now.